# Integrating augmented reality virtual patients into healthcare training: A scoping review of learning design and technical requirements

Farshid Chahartangi[1], Nahid Zarifsanaiey[2]*, Manoosh Mehrabi[2],
Bahareh Zeynalzadeh Ghoochani[3]

**1** Department of E-learning, Virtual School, Comprehensive Centre of Excellence for e-Learning in Medical Sciences, Student Research Committee, Shiraz University of Medical Sciences, Shiraz, Iran, **2** Department of E-learning, Virtual School, Comprehensive Centre of Excellence for e-Learning in Medical Sciences, Shiraz University of Medical Sciences, Shiraz, Iran, **3** Department of Occupational Therapy, School of Rehabilitation Sciences, Shiraz University of Medical Sciences, Shiraz, Iran

* nzarifsanaee@gmail.com

## Abstract

Augmented reality (AR) enables users to view the real world with enhanced digital information, making it a transformative tool in education. Virtual patients (VPs) technology is also defined as "a specific type of computer-based application that simulates real-world clinical scenario. AR and VPs offer interactive and immersive learning experiences, with AR enhancing the understanding of complex concepts and VPs providing hands-on practice in clinical scenario. This scoping review aims to identify an integrated learning design framework and the technical requirements for augmented reality-based virtual patient Simulation in healthcare professions education. This study employed a scoping review methodology that adhered to the PRISMA-ScR checklist and the Joanna Briggs Institute (JBI) guidelines, conducted between September and October 2024. The review covered six reputable databases: MEDLINE (PubMed), Science Direct (Elsevier), Web of Science (Clarivate), Cochrane library, ERIC, Scopus. A comprehensive search yielded 924 potential studies. Articles were selected via a two-stage screening process, involving title/abstract and full-text reviews based on predefined inclusion criteria. Disagreements were resolved through consultation, resulting in 27 studies being included. Eligible studies focused on augmented reality (AR)-based virtual patient (VP) technology in healthcare education, encompassing observational, quasi-experimental, and descriptive designs. Exclusions comprised grey literature, irrelevant studies, non-full-text articles, and non-AR/VP-focused research (e.g., standalone virtual reality). Various design approaches were employed, including situated learning, experiential learning, and the ADDEI model. The technical foundations of these studies were diverse, with Unity, UTTIME and PalpSim being commonly used software platforms. It is recommended that future studies thoroughly investigate each of these design framework and the

**Data availability statement:** All relevant data are within the paper and its Supporting Information files.

**Funding:** The author(s) received no specific funding for this work.

**Competing interests:** The authors have declared that no competing interests exist.

technical requirements of VPAR, examining them in greater detail from various cultural, economic, social, and emotional perspectives. Tackling these problems will be a crucial stride towards enhancing and optimizing education for healthcare professions education.

## Introduction

Recent advancements in digital technology have necessitated a re-evaluation of traditional teaching methodologies, particularly within healthcare education. The integration of innovative strategies that promote active student engagement has shown significant potential to enhance learning outcomes and prepare future healthcare professionals for evolving clinical demands [1] Among these strategies, the adoption of advanced digital tools—such as augmented reality (AR), mixed reality (MR), extended reality (XR), and virtual patient (VP) simulations—has gained considerable traction in educational settings worldwide [2].

Augmented reality, in particular, has emerged as a transformative tool in healthcare education. By overlaying digital information onto the physical world, AR creates immersive learning environments that enhance student interaction with educational content and simplify complex concepts [2]. Research indicates that AR positively influences learning outcomes, offering benefits such as improved knowledge retention, increased motivation, and enhanced performance accuracy [2]. Similarly, XR/AR simulators have demonstrated multiple advantages in healthcare training, including reduced errors, shortened simulation times, and support for cognitive-psychomotor tasks [3]. Despite these benefits, challenges such as technical complexities, educator resistance, and the need for streamlined implementation remain significant barriers to widespread adoption [3].

A prominent application of AR in healthcare education is the use of virtual patients (VPs). Defined by the Association of American Medical Colleges (AAMC) as "computer-based programs that simulate real-life clinical scenarios," VPs enable learners to assume the role of healthcare providers, practice clinical skills, and make diagnostic and treatment decisions in a controlled environment [4]. Over the past decade, the use of VPs has grown significantly, with research highlighting their effectiveness in improving diagnostic accuracy, clinical reasoning, and decision-making skills [5]. Virtual patient simulations also promote self-directed learning, active engagement, and timely feedback, making them a valuable tool for flexible and independent learning [6]. These simulations often incorporate 3D clinical settings, human physiology engines, and interactive avatars, further enhancing their realism and educational value [7,8].

While VP simulations provide a safe and controlled environment for practicing clinical skills, they often lack the dynamic and contextual elements inherent in real-world clinical encounters. Augmented reality, on the other hand, has the potential to bridge this gap by overlaying digital information onto the physical world, creating a more immersive and interactive learning experience [4]. The integration of VP simulations

with AR technologies offers a promising approach to healthcare education, combining the strengths of both tools to enhance learners' knowledge, skills, and confidence [9,10]. Studies suggest that this combination positively impacts learning performance, motivation, and engagement, while also supporting personalized learning and active knowledge acquisition [11–13].

Despite the growing interest in AR and VP technologies, the integration of virtual patient simulations with augmented reality remains underexplored. Most existing research has focused on the use of these technologies in isolation, with limited empirical evidence on their combined effectiveness in improving specific clinical outcomes, such as diagnostic accuracy, decision-making, and patient communication skills [13,14]. Furthermore, the optimal design features and educational approaches for integrating VP simulations with AR have yet to be fully defined, highlighting a critical gap in the literature.

Recognizing the potential of these technologies, our research team conducted a scoping review to synthesize existing evidence on the integration of virtual patient technologies and AR in healthcare education. This review aims to address gaps in comprehensive evaluations and provide an overview of the technical requirements, design considerations, and assessment methods for these immersive learning tools. By doing so, we seek to establish a conceptual framework for augmented reality-based virtual patient (VPAR) systems in healthcare education, guiding their effective development and implementation.

The primary objective of this scoping review is to conceptualize a comprehensive framework for the integration of augmented reality-based virtual patient (VPAR) systems in healthcare professions education.

### Specific goals

1. Identify an Integrated Learning Design Framework for the effective integration of VPAR systems into healthcare training curricula, including key pedagogical strategies, interaction mechanisms, and learning environment considerations.

2. Determine the Technical Requirements Framework for the development and implementation of VPAR systems, encompassing hardware, software, and supporting technologies necessary for seamless integration and optimal learning outcomes.

## Materials and methods

### Study design and framework

This scoping review was conducted from September to October 2024, following the Preferred Reporting Items for Systematic Reviews and Meta-Analyses Extension for Scoping Reviews (PRISMA-ScR) guidelines and the Joanna Briggs Institute (JBI) methodology for scoping reviews. The review was structured using the Population, Concept, and Context (PCC) framework to ensure a systematic and comprehensive approach [15].

**Eligibility criteria.** *Inclusion Criteria:*
Study Focus: Only studies integrating both augmented reality (AR) and virtual patient (VP) technologies in healthcare professions education were included. Studies utilizing only one technology (AR or VP alone) were excluded unless they explicitly combined both.

Study Designs: Observational (e.g., cohort studies), quasi-experimental (e.g., randomized controlled trials, pre-post studies), and descriptive (e.g., case studies, qualitative studies).

Language: English-language publications.

Timeframe: No date restrictions were applied; all available literature meeting the criteria was considered.

*Exclusion Criteria:*

• Grey literature, non-full-text articles, studies not directly addressing AR/VP integration, and publications from dubious databases.

- Studies focused solely on **virtual reality (VR)** without AR or VP components.

 **The PCC framework. Population:** Healthcare profession students (medical, nursing, allied health) engaged in AR-based VP educational interventions.

 **Concept:** Integration of AR and VP as educational tools, including:

Design, implementation, and evaluation of AR-based VP simulations.

Impact on learning outcomes (e.g., knowledge acquisition, clinical reasoning).

 **Context:** Healthcare education settings (undergraduate, postgraduate, continuing professional development).

- Population (participants): Healthcare profession students (e.g., medical, nursing, and…).

## Information sources and search strategy

A comprehensive search strategy was developed and executed across seven bibliographic databases: MEDLINE (PubMed), Science Direct (Elsevier), Web of Science (Clarivate), Cochrane Library (Embase and ClinicalTrials.gov), Scopus, and ERIC. The search strategy was designed in three stages:

- Preliminary Search: Two authors independently conducted a pilot search in MEDLINE (PubMed) to identify relevant keywords and refine the search strategy.

- Refinement: Search terms and strategies were compared, discussed, and refined until consensus was reached.

- Application: The final search strategy was applied to all seven databases.

 The search strategy included Boolean operators and key terms such as "augmented reality," "virtual patient," and "healthcare education." The full search strategy is detailed in Table 1.

## Study selection process

All identified records were imported into EndNote 21 for duplicate removal. Two independent reviewers screened titles and abstracts against the inclusion criteria. Full-text articles of potentially relevant studies were retrieved and assessed using the Rayyan system. Disagreements were resolved through discussion or consultation with a third reviewer. The study selection process is summarized in a PRISMA-ScR flow diagram (Fig 1).

**Table 1. Full search strategy in PubMed, conducted in October 2024.**

| Search | Query | Results |
|---|---|---|
| **#1** | ("design"[Title/Abstract]) AND ("virtual patient"[Title/Abstract]) OR ("patient, simulation*"[Title/Abstract]) OR ("simulate, patient*"[Title/Abstract]) | 1,704 |
| **#2** | (("augmented reality"[Title/Abstract]) OR ("mixed reality"[Title/Abstract])) OR ("extended reality"[Title/Abstract]) | 6,557 |
| **#1 AND #2** | ("design"[Title/Abstract]) AND ("virtual patient"[Title/Abstract]) OR ("patient, simulation*"[Title/Abstract]) OR ("simulate, patient*"[Title/Abstract] AND ("augmented realit*"[Title/Abstract]) OR ("mixed realit*"[Title/Abstract]) OR ("extended realit*"[Title/Abstract]) NOT ("virtual realit*"[Title/Abstract]) | 1,037 |
| **#3** | #1 AND #2 article type: Case Reports, Classical Article, Clinical Study, Clinical Trial, Clinical Trial Protocol, Controlled Clinical Trial, English Abstract, Guideline, Interactive Tutorial, Interview, Introductory Journal Article, Practice Guideline, Randomized Controlled Trial. | 99 |
| **#4** | #1 AND #2 Species: Humans | 69 |
| **#5** | #1 AND #2 Article language: English | 59 |
| **#6** | #1 AND #2 Text availability: Free full text | 23 |

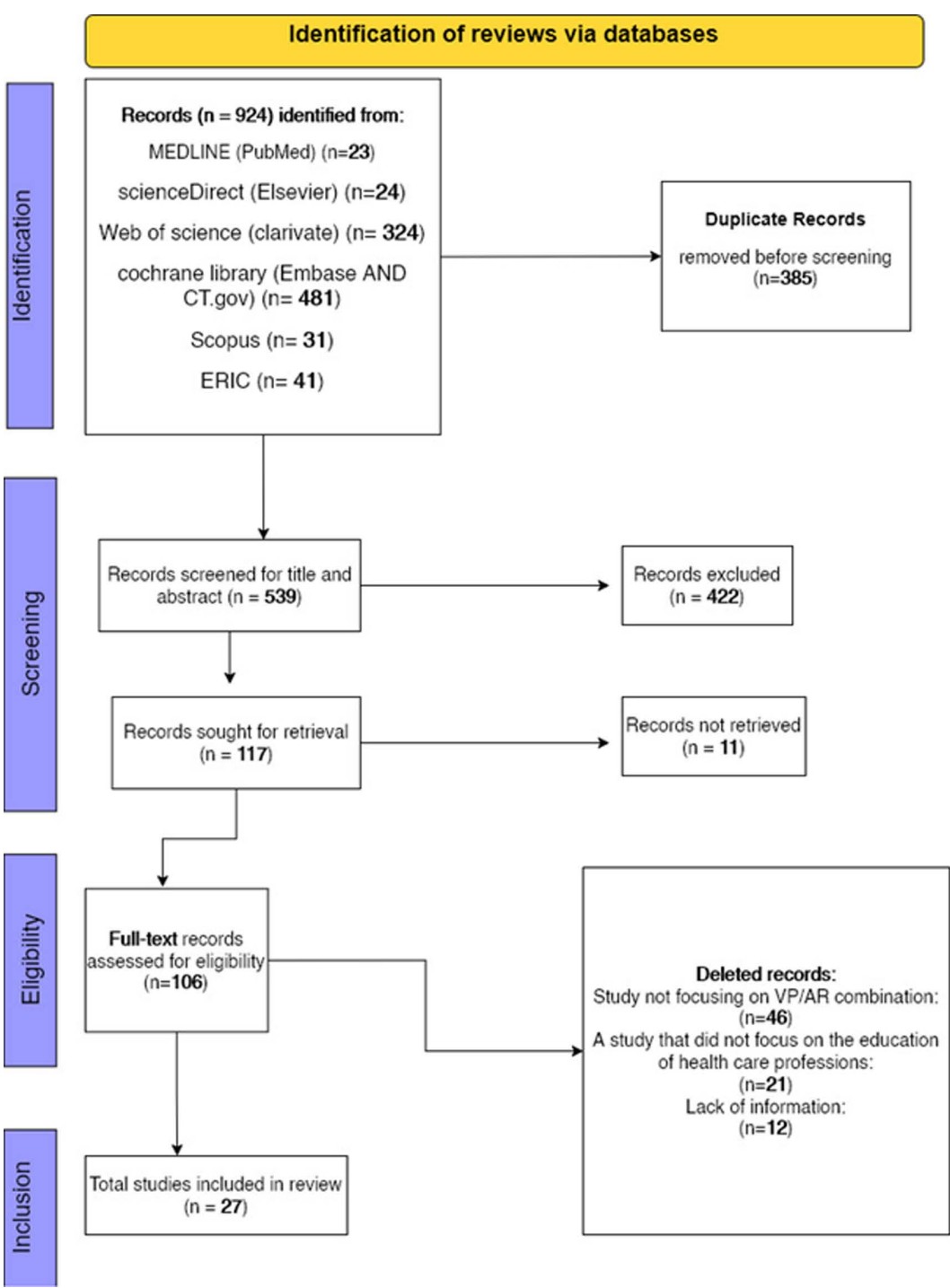

**Fig 1. PRISMA-SCR flow diagram of the search and study selection process for virtual patient and augmented reality integration in health care professions education.**

## Data extraction

A standardized data extraction form was developed and piloted. Two reviewers independently extracted data from the included studies, focusing on:

- Design methodologies of AR-based VP simulations.

- Evaluation techniques used to assess educational outcomes.

- Technical requirements for implementing AR and VP technologies.

- Outcomes related to knowledge acquisition, skill development, and learner engagement.

Discrepancies in data extraction were resolved through discussion or consultation with a third reviewer. The extracted data were organized into a structured table (Table 2).

## Data collection process

During the initial screening phase, two independent reviewers meticulously evaluated each study included in the scoping review on integrating augmented reality virtual patients (AR-VPs) into healthcare training. Each reviewer independently assessed the studies using predefined inclusion criteria, focusing on methodological rigor, relevance to the research questions, and alignment with the review's objectives. When discrepancies arose between the reviewers' assessments, they engaged in comprehensive discussions to reconcile differences, ultimately reaching a consensus through collaborative dialogue. This consensus-based approach ensured a systematic and objective evaluation, minimizing potential individual biases and enhancing the reliability of the study selection process.

To formally appraise the quality of the included studies, the reviewers employed the JBI (Joanna Briggs Institute) checklist, which provides a validated framework for assessing methodological quality in various research contexts. The detailed appraisal involved a systematic scoring process using this checklist, which evaluated multiple dimensions, including research design quality, methodological rigor, statistical significance, and potential **limitations. Reviewers thoroughly** documented their rationale for each scoring decision, ensuring a transparent and traceable assessment methodology.

## Quality assessment

To ensure a comprehensive evaluation of the studies included in this scoping review on AR-based virtual patients in healthcare training, two independent reviewers conducted a thorough quality assessment using the JBI checklist. Each reviewer systematically appraised the studies against predefined criteria, focusing on several key areas: methodological quality, relevance to learning design and technical requirements, and alignment with the objectives of the review.

To address any initial disagreements in assessments, the reviewers engaged in iterative discussions, ultimately reaching a consensus. This collaborative approach guaranteed an unbiased and standardized selection process.

A structured scoring system was utilized to evaluate critical aspects of each study, including:

- Research Design Validity: Assessing the robustness of the study's design.

- Technical Implementation Rigor: Evaluating the thoroughness of the technical execution.

- Educational Outcomes Measurement: Analyzing how effectively educational outcomes were measured.

- Reported Limitations: Reviewing the transparency and acknowledgment of limitations within each study.

The reviewers meticulously documented their scoring rationales to enhance transparency, creating an auditable trail for methodological decisions. This rigorous approach prioritized studies that effectively addressed the integration of AR virtual patients in healthcare education while upholding high scientific standards.

**Table 2. Summary of study characteristics and extracted data.**

| | Author. Year. Country. | Title | Type of Study. Place of Performance. | Sample Study. Level of Complexity of Simulation and Scenario. | Design Model | Tools Used. Health Care Uses and Scope. | Assessment Methods | Results | Limitation |
|---|---|---|---|---|---|---|---|---|---|
| 1 | d'Aiello, A. 2023. Italy [16]. | Impact of Holographic Heart Models and Mixed Reality for Learning Anatomy in Congenital Heart Diseases: An Exploratory Study | Experimental intervention with three groups. Institute of Congenital Heart Disease Center for Children and Adults (ACHD UNIT). | Medical students. Very high. | Experimental intervention with three groups | ARS-CADPT MR system, Interactive holographic glasses technology with hand head tracking and room spatial mapping. Improve knowledge, skills, and attitudes in congenital heart problems. | Questionnaire for recommend ability and usability, observation of the facilitator | No significant performance differences between mixed reality group and training video group | Few participants, limited to one class |
| 2 | George, O. 2023. United Kingdom [12]. | Augmented Reality in Medical Education: A Feasibility Study Using Mixed Methods | Single-group quasi-experimental study. UK university. | Medical students. High-fidelity simulation, high-complexity scenario. | Quasi-experimental | Microsoft HoloLens (2nd gen), Microsoft Remote Assist, Microsoft Teams, Microsoft PowerPoint. Improve skills, attitude in critical care (rapid sequence intubation). | Valid Adapted Immersive Experience Questionnaire (AIEQ), Summary Intrinsic Motivation Questionnaire (AIMI), Group interview | Remote simulation acceptable, high enjoyment and value | Few participants, limited to one class |
| 3 | Baratz, G. 2022. United States [17]. | Comparing learning retention in medical students using mixed-reality to supplement dissection: a preliminary study. | A preliminary study (a balanced design model and a non-randomized trial). Case Western Reserve University, School of Medicine. | First year medical student. Medium fidelity simulation level, complex scenario level. | Part 1 contains a balanced design. Part 2 is a non-randomized controlled trial conducted 8 months after Part 1. Augmented reality technology, mixed reality technology, corpse for dissection. | Computer and monitor user interface. Knowledge improvement Learning female breast anatomy with augmented and mixed reality. | Test. Survey. Questionnaire. | This study suggests that medical students prefer mixed reality for learning breast anatomy and that it enhances teamwork. It also indicates that using mixed reality alongside cadaver autopsy can improve long-term retention of anatomical knowledge. | The newness of MR may influence participants' satisfaction with the method. The small sample size of the intervention and the small number of quiz questions may also cause bias. |
| 4 | Stone, N. 2022. United States [18]. | Remote Surgery Training Using Synthetic Models with Augmented Reality Headsets | Initial test. Ikan Faculty of Medicine. | Distance education of doctors. High fidelity, complex scenario. | Initial test | Web RTC platform, mpMRI AR headset, and HIPPA and GPDR compliant software. Skill improvement in fusion prostate biopsy guided by mpMRI. | Educational content survey, user feedback | Distance learning platform successfully beta tested, feasible and practical | Small sample size for phantom and headset testing |

*(Continued)*

**Table 2.** (Continued)

| | Author. Year. Country. | Title | Type of Study. Place of Performance. | Sample Study. Level of Complexity of Simulation and Scenario. | Design Model | Tools Used. Health Care Uses and Scope. | Assessment Methods | Results | Limitation |
|---|---|---|---|---|---|---|---|---|---|
| 5 | Hess, O. 2022. United States [19]. | Training Communication Skills Using Remote Medical Simulation with Augmented Reality: A Qualitative Feasibility and Acceptability Study | Descriptive (three-group intervention) and qualitative (phenomenological) study. Stanford School of Medicine. | Second year medical students and pre-clinical doctors. High fidelity, complex clinical scenario. | Descriptive and qualitative study | CHARM Simulation, Magic Leap One (ML1) headset. Improve knowledge, skills, and attitudes in cardiac arrest medical crisis. | REDCap electronic questionnaire, qualitative research with focus group interview | Positive experiences with CHARM Simulation, effective social distance learning | Disproportion of avatars, high cost of AR headsets |
| 6 | Herbert, V. 2021. United States [20]. | Development of a Smartphone Application with Augmented Reality to Support the Virtual Learning of Nursing Students in the Field of Heart Failure | Quasi-experimental design with pretest-posttest. Maine School of Nursing Orono. | Second semester nursing students. Medium fidelity, complex scenario. | Quasi-experimental design | AR camera, mobile phone, Apple's ARKit 2. Knowledge improvement in anatomy and physiology of the heart (heart failure). | Pre-test-post-test heart failure assessment (HFA) | AR applications provide extensive information, should be combined with other teaching methods | Reliability and validity data inconsistent, small sample size |
| 7 | Zackoff, M. 2021. United States [21]. | Development and Implementation of High-Fidelity Augmented Reality Simulation for Patient Compensation Diagnosis | Prospective observational study. Grand Children's Hospital. | Medical students. High fidelity, complex scenario. | Prospective observational study | AR dummy, ZBrush, Unity platform. Skill improvement in mental status, respiratory status, perfusion status of patient. | Checklist, Qualtrics online survey | AR increases realism of mannequin simulation, improves diagnosis | Limited to intensive care units, small study size |
| 8 | Caruso, T. 2021. United States [22]. | Integrated eye tracking on Magic Leap One during augmented reality medical simulation: a technical report. | A technical report. University Hospital in Northern California. | A convenience sample of adults aged 18 and over was selected. High fidelity simulation level and complex scenario level. | The aim of this project is to improve the look tracking capabilities of the CHARM Simulation It was the Magic Leap One (ML1). | Headset, bed and monitor, holographic touch pads. Improving knowledge and skills Eye tracking and vital signs. | questionnaire | The study was concluded after the final three participants achieved gaze acquisition rates of 80, 80, and 80.1 percent, respectively, indicating that eye tracking technology can be reliably used with ML1 enabled with CHARM Simulation software. | not mentioned |

*(Continued)*

Table 2. (Continued)

| | Author. Year. Country. | Title | Type of Study. Place of Performance. | Sample Study. Level of Complexity of Simulation and Scenario. | Design Model | Tools Used. Health Care Uses and Scope. | Assessment Methods | Results | Limitation |
|---|---|---|---|---|---|---|---|---|---|
| 9 | Mishvelov, A. 2021. Russia [23]. | Computer-assisted surgery: virtual and augmented reality displays for navigation during planning and performing surgery on large joints | Development of software and equipment of a prototype. Stavropol Regional Clinical Hospital. | Doctor specializing in surgery. High fidelity simulation level, complex scenario level. | An analog for processing CT and MRI data, the Vitrea2 workstation is a hardware and software suite. DICOM images, the HoloWiver module, displays medical data on HoloLens glasses. | Camera, HoloLens glasses, software. Improving knowledge and skills Surgery on the human skeleton and lumbar organs (surgery on the kneecap). | Observation and personal opinion of surgical specialists. | This technique allows you to perform surgical interventions both individually and in groups at different levels of training without the need for expensive models. | That the number of surgeries and organs were limited. There were no proper evaluation methods. The cost of HoloLens glasses was high. There was no equal method of comparison. |
| 10 | Amiras, D. 2021. United Kingdom [24]. | Augmented reality Simulation for CT-guided interventions | Development and evaluation of augmented reality systems (feasibility study). Imperial University London. | Senior interventional radiologists and trainee radiologists. High fidelity simulation, complex scenario level. | The virtual patient was created using a CT dataset obtained from the Cancer Imaging Archive. | Dummy phantom, ChArUco indicators, mannequin, monitor, halo lens. Skill improvement Radiology with ct. | Questionnaire with Likert scale | suggested that the developed Simulation could be an effective training tool for clinical practice skills | The limitation of the present study is the fact that the operation time and radiation dose were not calculated. |
| 11 | Liu, S. 2021. China [25]. | A 3D hologram with mixed reality techniques to improve understanding of lung lesions caused by COVID-19: a randomized controlled trial | Randomized controlled trial. Wuhan University and Hospital. | 20 radiologist students, 20 surgeon students and 20 medical students (each was divided into two groups). High fidelity simulation level, complex scenario level. | The CT scan data was performed using a workstation (Star Cloud) of a 3D reconstruction software from (Visual3D) mixed reality. | Monitor, CT images, software, Microsoft Halogens. Improving knowledge, skills and attitudes. Pulmonary lesions. | Activity efficiency. Questionnaire. The National Aeronautics and Space Administration Task Load Index (NASA-TLX) was used to assess the participants' mental workload. | A 3D hologram with mixed reality techniques could be used to improve medical professionals' understanding of lung lesions caused by COVID-19. Can It has been used in medical education to improve spatial awareness, increase interest, improve comprehension and reduce the learning curve. | Universities face financial, ethical, and regulatory challenges with cadaver samples, and the COVID-19 pandemic increases the risk of virus transmission. |

*(Continued)*

| | Author. Year. Country. | Title | Type of Study. Place of Performance. | Sample Study. Level of Complexity of Simulation and Scenario. | Design Model | Tools Used. Health Care Uses and Scope. | Assessment Methods | Results | Limitation |
|---|---|---|---|---|---|---|---|---|---|
| 12 | Djenouri Y. 2021 Norway [26]. | Secure Collaborative Augmented Reality Framework for Biomedical Informatics | A technical report. University. | Helping doctors. High fidelity, complex clinical scenario. | In the SCF-BHI framework, AR enhances the visualization of medical patterns to support accurate clinical decision-making. Integrated with HoloLens, the system overlays deep learning outputs—such as brain tumor regions—onto real-world views, providing clear, interactive visualization for physicians. | ARAM Library, HoloLens, Deep Learning. brain tumor regions | a combination of quantitative metrics (IoU, runtime, percentage of detected attacks) and comparative analyses to thoroughly evaluate the performance of their proposed SCF-BHI framework | The SCF-BHI framework integrates deep learning, multi-agent systems, augmented reality, and block chain for biomedical health informatics, improving medical data visualization and collaboration security. It showed effectiveness in biomedical segmentation. | Future work could focus on enhancing deep learning models, optimizing real-time processing, and expanding the framework to other medical applications like tumor detection and disease diagnosis. |
| 13 | Mladenovic, R. 2020. Serbia [27]. | Effect of augmented reality simulation on local anesthesia administration | Prospective study (intervention with study group and control group). University of Pristina, Faculty of Dentistry. | 4th and 5th year dental students. High fidelity simulation, complex scenario level. | Mobile Simulation on AR supported device. Augmented reality mode with Vuforia engine (PTC) using the camera of a mobile device or virtual reality glasses. | Plastic model, mobile phone, camera, virtual reality glasses. Skill improvement. Local anesthesia in pediatric patients (dentistry). | Saliva samples before and after anesthesia were measured as one of the indicators of acute stress by Sarstedt Salivette | The AR concept may contribute to better syringe manipulation and control in students administering their first anesthetic injection to pediatric patients, but may not reduce acute stress. | It was not mentioned |
| 14 | Sushereba, C. 2020. United States [28]. | Virtual patient immersive trainer to train perceptual skills using augmented reality | Compilation and development of an educational software. University. | Doctors and medical students. High fidelity simulation level, complex scenario level. | In VPIT training, the student views the virtual patient through an AR headset, while the trainer uses a tablet to select and guide the simulation scenario, tracking the student's focus throughout. | Tablet, training mannequin, AR headset. Knowledge improvement. Critical patient assessment. | The relevant professor's opinion about the level of comprehensive knowledge Coach's comment | VPIT is portable, easy to set up, and doesn't require special training, making it ideal for simulation training in healthcare settings without dedicated simulation centers, such as small or rural facilities. | Medical training mannequins are limited by their rare availability, inability to quickly change appearance, and the need for a specific number of people in a university setting. |

*(Continued)*

| | Author. Year. Country. | Title | Type of Study. Place of Performance. | Sample Study. Level of Complexity of Simulation and Scenario. | Design Model | Tools Used. Health Care Uses and Scope. | Assessment Methods | Results | Limitation |
|---|---|---|---|---|---|---|---|---|---|
| 15 | Aebersold, M. 2018. United States [29]. | Interactive anatomy-augmented virtual simulation training | This combined study (quantitative: control group and study group. Qualitative: survey). Great Midwestern institution. | Second year nursing students. High fidelity simulation level, complex scenario level. | Simulation-based education (SBE). Video and educational content (nasogastric intubation) based on the institution's standard curriculum (combines video with 3D computer graphics). | IPad using AR, mobile device, computer, video. Piping. Skill improvement. Nasogastric tube (NGT) placement. | Checklist of NGT Qualtrics online survey Data collection with Mann-Whitney test, independent t, chi-score, Fisher | The AR module was well received by participants, with 86% of the AR group rating it superior or much superior to other procedural training programs they had experienced. | This was a small study done in a school with students who were already exposed to this skill |
| 16 | Nuanmeesri, S. 2018. Thailand [30]. | The Augmented Reality for Teaching Thai Students about the Human Heart | Development of tools with experimental design before and after the intervention. Bangkok Rajabhat University | Students. Low fidelity simulation level and easy scenario level. | Augmented reality through smartphones or tablets with a scanning program to scan heart images. | Mobile phone, camera to scan images. Knowledge improvement. Teaching heart anatomy (human heart function, heart components, and heartbeat and blood circulation). | Pre-test and post-test grades To evaluate the technology from the integrated theory of acceptance and use of technology (UTAUT). From the objective conformity index (IOC), diffusion of innovation (DOI) and content validity index (CVI). | The development of augmented reality as a tool for teaching about the human heart can contribute to effective learning and better outcomes in understanding. | It was not mentioned |
| 17 | Zielke, M. 2017. United States [10]. | Development of virtual patients with VR/AR for a natural user interface in medical education | Technology prototype. Medical School. | Medical students. Low fidelity simulation level, medium scenario. | UT TIME portal platform which is web based through laptop or desktop computer. Text-based platform interface and monitor. | Oculus is a VR head-mounted display device, Microsoft HoloLens AR. Improving attitude. Patient interview. | Survey with qualitative data | VR/AR technologies may advance the field of NUI research and provide the potential for interactive and customizable 3D virtual patients to practice effective communication skills in medical education. | not mentioned |
| 18 | Daher, S. 2017. United States [31]. | Optical See-Through vs. Spatial Augmented Reality Simulators for Medical Applications | Technology prototype. University of Central Florida. | Nursing and medical students. High fidelity simulation level, complex scenario. | Spatial Augmented Reality (SAR) uses multiple projectors and infrared cameras to display images on a surface. It tracks finger touches on the skin and sends the input to the simulation to determine the patient's response. | Physical mannequin, projector, infrared camera, touch tracking. Improve communication skills. Social interaction with the patient and attention to the patient's face was more. (Stroke test). | Survey questionnaire | It can be concluded that this study was conducted with the aim of comparing dynamic images between spatial augmented reality and optical vision augmented reality Simulation and their effects on depth perception, task completion and social presence. | This study was not well evaluated The number of participants was limited The tools used were very expensive. |

*(Continued)*

**Table 2.** (Continued)

| | Author. Year. Country. | Title | Type of Study. Place of Performance. | Sample Study. Level of Complexity of Simulation and Scenario. | Design Model | Tools Used. Health Care Uses and Scope. | Assessment Methods | Results | Limitation |
|---|---|---|---|---|---|---|---|---|---|
| 19 | Léger, É. 2017. Canada [32]. | Quantification of Attention Shift in Augmented Reality Image Guided Neurosurgery | Recommendation-analytical (comparative). Concordia University. | The subjects were master's students, researchers, engineers and neurosurgery assistants. High fidelity simulation level, complex scenario. | Image-guided surgery (IGS). Augmented reality views in IGS. | Tablets, projectors, surgical micro-scopes, semi-silvered mirrors, Head-mounted displays (HMD) or the use of the IGS system monitor. Skill improvement Brain surgery (tumor in the skull). | A subject matter expert's look at Manito and time calculation in both methods. questionnaire | The results show that tumor delineation with AR systems takes less time and requires less attention shifting than a traditional navigation system. It is not clear that mobile AR performs better than desktop AR in these two factors, but it is clear that users find it more intuitive and convenient. | The two most common nega-tive comments we received about our system were that there was some lag in the video feed and that the small screen size made it difficult to be accurate. |
| 20 | Vaughn, J. 2016. United States [33]. | Piloting aug-mented reality technology to enhance realism in clinical simulation | Experimental study. University of Dhaka, Faculty of Nursing. | Nursing students. High fidelity simulation level and complex scenario level. | It combines mannequin-based simulation with Google Glass, a head-wearable device that projects video into the students' field of vision. | YouTube, Adamak, augmented reality headset, Google glasses, software. Skill improvement. Acute asthma. | Survey using two accredited health care simulation experts. Feasibility assessment using a questionnaire | Improving visual realism in simula-tions is the first step toward enhancing realism. Adding AR technology to advanced simula-tions will transform immersive learning, making scenarios more realistic and deepen-ing students' understanding. | It provides enough fidelity for beginner learners but does not have many effects for advanced learners. The cost of the headset is high. Specialists and experienced people are limited. |
| 21 | Ntourakis, D. 2016. France [34]. | Augmented Real-ity Guidance for the Resection of Missed Colorectal Liver Metasta-ses: An Initial Experience. | Prospective pilot study. Strasbourg Hospital. | Physicians. High fidel-ity, complex scenario. | Prospective pilot study. | 3D virtual anatomical model, VR RENDER software, IRCAD AR software. Skill development, rare medical cases in colorectal cancer liver metastases (CRLM). | Qualitative data. | AR provides real-time anatomic cartography of liver, aids in oncologic resection of missed CRLM. | Image registration requires human interaction, deformation compensation. |

**Table 2.** (Continued)

| | Author. Year. Country. | Title | Type of Study. Place of Performance. | Sample Study. Level of Complexity of Simulation and Scenario. | Design Model | Tools Used. Health Care Uses and Scope. | Assessment Methods | Results | Limitation |
|---|---|---|---|---|---|---|---|---|---|
| 22 | Ma, M. 2016. Germany [35]. | Personalized Augmented Reality for Anatomy Education | Survey research. University of Munich, Germany. | Seven doctors and 72 anatomy students. High fidelity simulation level, complex scenario level. | An overview of the system, which includes a large display device and a Microsoft Kinect. It uses the magic mirror concept for personal visualization of organs (CT). Natural personal interactions and play with gesture-based interaction. | Kinect device, body temperature, camera. Knowledge promotion For the important bones and organs of the chest and abdomen. | Questionnaire with Likert scale Expert judgment | Our system is certified accurate and useful enough for learning anatomy, and most medical students embraced the learning potential of this technology | not mentioned |
| 23 | Nifakos, S. 2014. Sweden [36]. | Virtual patients in a real clinical context using augmented reality: Impact on antibiotics prescription behaviors | Combined method (model method, design method). Department of Medical Simulation at Karolinska University Hospital. | Healthcare professionals with 2 years of prescribing experience. Medium fidelity simulation, complex scenario. | Metaio AR platform for developing digital artifacts and learning context tracking (virtual patient integration and augmented reality) built on a JavaScript link in combination with XML content. | Tablet, web, augmented reality tracker. Knowledge improvement Pharmacy (prescribing). | Employee survey | VpAR is the result of an iterative design-based approach in which practitioners were involved and combines the best of both approaches. | It is necessary to use a tablet device. As AR technology improves, the findings of this study will be prototyped in new solutions (e.g., wearable devices, Google Glasses) to improve the usability of the interaction. |
| 24 | González, F. 2014. Mexico [37]. | Smart multi-level tool for remote patient monitoring based on a wireless sensor network and mobile augmented reality | Development of SMTRPM technology. Kyodad Giuriz University. | Nurse activities. High fidelity simulation level, complex scenario. | A system for remote monitoring of patient's body temperature and heart rate using a wireless sensor network (WSN) and mobile augmented reality (MAR). | Arduino microcontrollers, PCs, smartphones, sensors, WiFly technology, and software like LabVIEW. Knowledge, skill, attitude Body temperature and heart rate of the patient. | The efficiency of the system was determined by its ability to accurately measure body temperature and heart rate in real time during patient care | This study demonstrates the development and successful evaluation of a system that enables remote monitoring of body temperature and heart rate using a wireless sensor network and mobile augmented reality technology. | not mentioned |

*(Continued)*

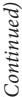

| | Author. Year. Country. | Title | Type of Study. Place of Performance. | Sample Study. Level of Complexity of Simulation and Scenario. | Design Model | Tools Used. Health Care Uses and Scope. | Assessment Methods | Results | Limitation |
|---|---|---|---|---|---|---|---|---|---|
| 25 | Luciano, C. J. 2011. United States [13]. | Learning retention of thoracic pedicle screw placement using a high-resolution augmented reality Simulation with haptic feedback | Software development. Bangar University, Faculty of Computer Science. | Department of Radiology. High fidelity simulation level, complex scenario. | A new augmented reality simulation called PalpSim has been developed that allows trainees to feel a virtual patient using their hands. | Dual force feedback devices, PHAN-TOM Omni needle interface, chroma-ke touch hardware. Skill improvement Femoral palpation and needle insertion training. | Experts' opinion about the software | This simulation offers a high level of facial fidelity and is one of the first medical simulation devices to integrate touch with augmented reality | not mentioned |
| 26 | Coles, T. R. 2011. Italy [38]. | Integrating Haptics with Augmented Reality in a Femoral Palpation and Needle Insertion Training Simulation | A pilot study. Chicago School of Engineering and Medicine. | Neurosurgeon resident. High fidelity simulation level, complex scenario. | ImmersiveTouch software uses a set of modules to collect, process, and present graphics and touch data, which are then seamlessly integrated onto the hardware platform. A virtual 3D volume of the human spine using CT from a single patient | Electromagnetic sensor attached to stereoscopic glasses, six-dimensional sensor located inside SpaceGrips (LaserAid). Improving knowledge and skills Thoracic stem screw placement. | questionnaire | Our simulation is adaptable for various training scenarios, with or without image guidance, and allows parameter adjustments based on teacher or trainee preferences, making it suitable for many surgical tasks. | Our study design was limited by data collection conditions |
| 27 | Lamounier, E. 2010. Brazil [39]. | On the use of Augmented Reality techniques in learning and interpretation of cardiologic data | Software development. University of Oberland, Faculty of Electrical and Computer Engineering. | Medical students. Medium fidelity simulation level, complex scenario. | The system processes traditional cardiac media into an intuitive format, using sensors for real-time data, cameras for visual recording, and markers for integration, blending the real and virtual worlds in visualization. | Camera, software, computer system, sensor, internet. Improving knowledge and skills Cardiovascular. | The system's effectiveness can be measured by its ability to process cardiovascular data, support editing and visualization, and realistically simulate the heartbeat. | The system uses markers to synchronize the real and virtual worlds, allowing the integration of real-world scenes captured by the camera with virtual objects such as hearts and charts. | not mentioned |

## Collating and summarizing data

Following the quality assessment phase, the data were systematically categorized and analyzed. A summary table was created to outline the characteristics and findings of the included articles, and a comprehensive list of studies was compiled. An overview of the studies was conducted by systematically analyzing their geographic distribution, publication years, outcomes, and content to identify the benefits, effects, and challenges associated with the study's objectives.

## Results

### Included reviews

A comprehensive search strategy was implemented across multiple databases from September 25 to October 15, 2024, yielding a total of 924 records. The distribution of records across databases was as follows: MEDLINE (PubMed) (n = 23), Science Direct (Elsevier) (n = 24), Web of Science (Clarivate) (n = 324), Cochrane Library (Embase and ClinicalTrials. gov) (n = 481), Scopus (n = 31), and ERIC (n = 41). After removing 385 duplicate records, 539 entries underwent title and abstract screening, resulting in the exclusion of 422 records. The remaining 117 records were retrieved for full-text evaluation. Following full-text analysis, 79 articles were excluded due to the following reasons:

• No focus on the integration of virtual patients (VPs) and augmented reality (AR) (n = 46).

• No focus on healthcare professions education (n = 21).

• Insufficient information to determine eligibility (n = 12).

Ultimately, 27 studies were deemed relevant and included in this review. The study selection process is summarized in the PRISMA-ScR flow diagram (Fig 1).

A detailed overview of the studies incorporated in this review as offered in Table 2.

### Characteristics of included studies

**Geographical distribution.** The 27 included studies originated from diverse geographical regions, with the majority (55.56%, n = 15) from the Americas, followed by Europe (37.04%, n = 10), and Asia (7.41%, n = 2). Notably, Africa was not represented in the reviewed studies. Within the Americas, the United States contributed the most studies (n = 12), followed by Brazil, Mexico, and Canada (n = 1 each). In Europe, the United Kingdom led with 2 studies (n = 2), followed by Italy (n = 2), and France, Russia, Serbia, Norway, Germany, and Sweden (n = 1 each). The limited representation from Asia included studies from China and Thailand (n = 1 each) (Fig 2).

**Participant demographics.** The participants in the reviewed studies were predominantly medical students (40.74%, n = 11), followed by nursing students (18.52%, n = 5), radiology students (11.11%, n = 3), and preclinical and other students (11.11%, n = 3). Other groups included medical residents (7.41%, n = 2), dental students (3.70%, n = 1), pharmacy students (3.70%, n = 1), and medical Informatics (3.70%, n = 1) (Fig 3).

### Medical SPECIALTIES AND EDUCATIONAL APPLICATIONS

The included studies (n = 27) showcased the application of augmented reality-based virtual patient (AR-VP) technology across various medical specialties. Two reviewers independently extracted and classified the data according to the primary educational focus of each study, resolving discrepancies through discussion and consulting a third reviewer when necessary. The classifications were derived inductively from study objectives and outcomes, without utilizing qualitative analysis software.

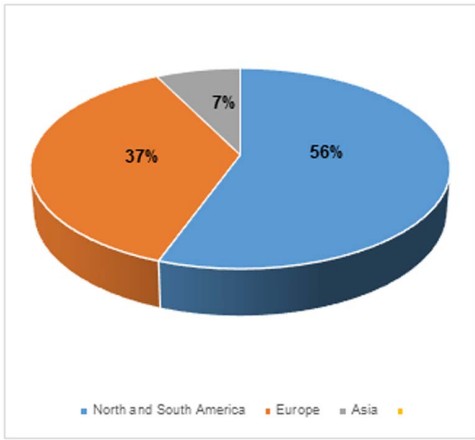

**Fig 2. Geographical distribution chart of studies.**

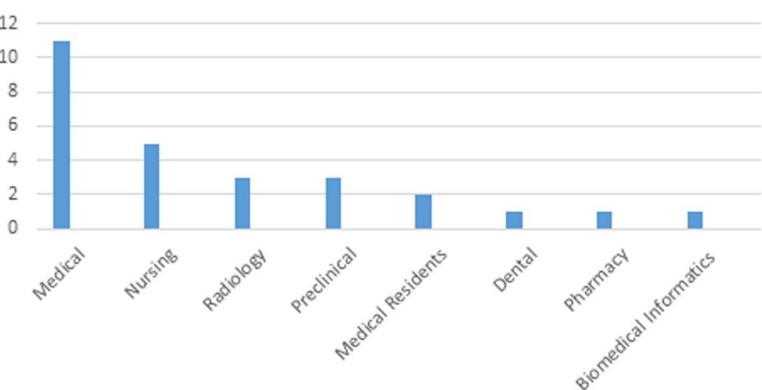

**Fig 3. Diagram of participants (students) in studies.**

*The major themes and applications identified included:*

- Advanced Clinical Procedures: Examples include mpMRI-guided prostate biopsies, emergency cardiac arrest management, and treatment planning for colorectal cancer metastases.

- Anatomical and Physiological Education: Focus areas encompass female breast anatomy, lower limb structures, skeletal system, and congenital heart abnormalities, alongside the use of CT scans for pulmonary lesion assessment and heart physiology.

- Diagnostic and Therapeutic Training: This includes diagnostic skills such as stroke diagnosis and critical patient assessment, as well as therapeutic skills like brain tumor management, prescription writing, and patient communication. Stroke diagnosis was classified under "therapeutic skills" when studies emphasized treatment decision-making.

- Practical Clinical Skills: Hands-on training involved vital sign monitoring, chest screw placement, and heart rate interpretation.

These studies underscore the importance of comprehensive training across medical specialties and the critical role of immersive technologies in developing essential clinical skills.

**Overview of medical contexts.**

• The reviewed studies covered a wide range of medical specialties and educational contexts, including:

• Advanced Procedures: mpMRI-guided prostate biopsies, emergency cardiac arrest responses, and colorectal cancer management.

• Anatomical Education: Female breast, lower limb, and skeletal anatomy.

• Diagnostic Training: Heart anatomy, physiology, pulmonary lesions, and radiological techniques (e.g., CT scans).

• Therapeutic Skills: Stroke diagnosis, brain tumor management, and patient communication.

• Practical Skills: Heart rate monitoring, chest screw placement, prescription writing, and vital signs management.

## Distribution of study designs

The chart illustrates the distribution of study designs among the 27 included studies, categorized into four main types: sample-experimental, descriptive-analytical, experimental, and quasi-experimental (Fig 4).

As we can see, the chart illustrates the distribution of study designs among the 27 included studies, categorized into four types: sample-experimental, descriptive-analytical, experimental, and quasi-experimental. Sample-experimental studies, representing 48% (n = 13) of the studies, focus on evaluating the initial effectiveness and feasibility of AR and VP technologies in educational settings. Descriptive-analytical studies, accounting for 22% (n = 6), analyze user interactions with these technologies to understand their advantages and challenges. Experimental studies, also at 22% (n = 6), employ controlled experiments to assess the direct impact of AR and VP on learning outcomes. Quasi-experimental studies, the smallest group at 8% (n = 2), explore initial impacts in specific educational contexts, providing preliminary insights for further research.

## Data analysis

**Specific Objective 1:** Identification of an Integrated Learning Design Framework for the use of virtual patients and augmented reality in healthcare professions education (Table 3).

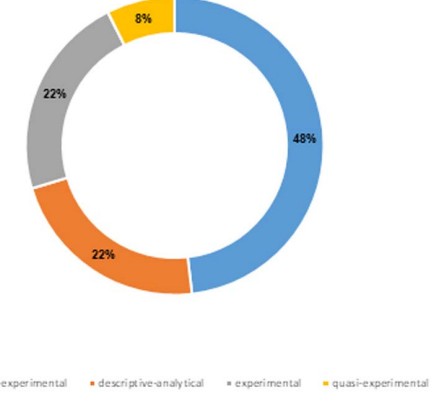

legend: ■ sample-experimental  ■ descriptive-analytical  ■ experimental  ■ quasi-experimental

**Fig 4. Chart of study types.**

**Table 3. Components of integrated learning design framework.**

| Components | Items | Description | Studies |
|---|---|---|---|
| features of VP/AR design models | simulation level | High fidelity level | 22 |
| | | Medium fidelity level | 3 |
| | | Low fidelity level | 2 |
| | Scenario level | simple | 3 |
| | | complicated | 24 |
| | Type of content | Photo, sound, video, text, 3D images, avatar, hyperlink, webinar (combining multiple contents at the same time in each simulation). | 27 |
| | teaching method | Refer to student-centered, professor-centered, and a combination of both. | 27 |
| **Educational strategies** | Problem-based approach | Studies that have used one or more problems to teach a subject. | 8 |
| | Prototype-based approach | Studies that for the first time have used new directions that usually use multimedia principles. | 7 |
| | Situational learning approach | Authentic and contextual learning is made possible by embedding educational experiences in the real world environment and by bringing the real world into the classroom. | 27 |
| | Game-based approach | Stories are about clinical cases and their activities. Studies that use several different methods to design scenarios that can refer to lectures, textual information, animation, images, etc. | 12 |
| **Interaction And feedback** | order | Where someone issues instructions to a system. This can be done in a variety of ways, including typing commands, selecting options from menus in a windowed environment or on a multi-touch screen, speaking commands aloud, pointing, pressing buttons, or using a combination of function keys. | 6 |
| | Conversation | Where a person interacts with a system. They can speak through an interface or type questions that the system answers through text or speech output. | 2 |
| | Manipulation | Where people interact with objects in a virtual or physical space by manipulating them (e.g., opening, holding, closing and placing). They can reinforce their familiar knowledge of how to interact with objects. | 7 |
| | explore | Where people move in a virtual environment or a physical space. Virtual environments include 3D worlds including augmented or virtual reality. They enable users to reinforce their familiar knowledge by physically moving around. Physical spaces that use sensor-based technologies. | 10 |
| | answering | Where the system initiates the interaction and the individual chooses whether to respond. For example, location-based mobile technology can inform people about points of interest. They can choose to look at or ignore the information that appears on their phone. | 2 |
| | diagnostic | This type of assessment supports learners in determining their current level of knowledge or capacity for a subject and ensures that misunderstandings are clarified before learning is delivered. | 2 |
| | formation | Information that is communicated to the learner and is intended to modify their thinking or behavior to improve learning. | 9 |
| | Summative | The information that has been transferred to the learner and is intended to measure the change in his behavior at the end of the course. | 16 |
| **Learning environments** | Hospital | The study implementation environment is in some of the hospital's departments | 8 |
| | University | The study implementation environment is in the classes or clinical departments of the university | 17 |
| | Private institutions | The study implementation environment in private institutions that are interested in technologies and Simulation | 2 |

*(Continued)*

**Table 3.** (Continued)

| Components | Items | Description | Studies |
|---|---|---|---|
| **Assessing learning outcomes** | Knowledge | It refers to teaching and permanence of the material. For example: learning the anatomy of human body parts, medical education, communication skills education, heart failure education, patient vital status education, prescription writing education. | 5 |
| | Skill | It refers to learning practical activities and practical practice and repetition. For example: congenital heart, rapid sequence intubation, colorectal cancer liver metastases, mpMRI-guided prostate fusion biopsy, and a cardiac arrest medical crisis. | 12 |
| | attitude | It refers to self-confidence and motivation and maintaining effective communication. For example: patient interview, diagnosis of vital signs, patient's mental state | 2 |
| | Composition | Skill was used. For example: surgery on the human skeleton and lower back organs, lung lesions, chest screw placement, rapid sequence intubation, patient's body temperature and heart rate. | 8 |
| | Objective measurement | Questionnaire for recommend ability and usability and observation, Valid Adapted Immersive Experience Questionnaire (AIEQ) and Intrinsic Motivation Summary Questionnaire (AIMI), REDCap electronic questionnaire, | 9 |
| | Subjective assessment and qualitative analysis | Qualtrics online survey, NGT checklist, survey using two accredited health care simulation experts, employee survey, and subject matter expert opinion with software. Interview, observation. | 11 |
| | Usability evaluation techniques | Assessment of heart failure (HFA), saliva samples before and after anesthesia as one of the indicators of acute stress by Sarstedt Salivette, task load index of National Aeronautics and Space Administration (NASA-TLX) to evaluate mental workload, integrated acceptance theory and Use of technology (UTAUT) and objective conformity index (IOC) and dissemination of innovation (DOI) and content validity index (CVI). | 7 |

The study identified five key components of an integrated learning design framework for VP & AR technologies in healthcare education. These components include VP/AR design models, educational strategies, interaction and feedback, learning environments, and learning outcomes evaluation.

*VP/AR Design Models:*

- Simulation fidelity: High fidelity (n = 22), medium fidelity (n = 3), and low fidelity (n = 2).

- Scenario complexity: Complex scenarios (n = 24) and simple scenarios (n = 3).

- Content types: Multimedia content, including photos, videos, 3D images, avatars, and hyperlinks.

- Teaching methods: Student-centered, instructor-centered, and blended approaches.

*Educational Strategies:*

- Problem-based learning: Engaging students with real-world problems (n = 8).

- Prototype-based learning: Focusing on innovative methods and technologies (n = 7).

- Situational learning: Utilizing real-world settings for practical experience (n = 27).

- Game-based learning: Incorporating clinical cases and interactive activities (n = 12).

*Interaction and Feedback:*

- Interaction types: Commanding, conversing, manipulating, exploring, and responding.

- Feedback types: Diagnostic, formative, and summative feedback to enhance learning outcomes.

*Learning Environments:*

- Hospitals: Clinical settings for hands-on training (n = 8).

- Universities: Academic and clinical departments (n = 17).

- Private institutions: Technology-focused training centers (n = 2).

*Learning Outcomes Evaluation:*

- Knowledge: Retention and understanding of medical concepts (n = 5).

- Skills: Practical application and repetition (n = 12).

- Attitudes: Confidence, motivation, and communication (n = 2).

- Assessment methods: Objective measurements (n = 9) and subjective assessments (n = 11).

Overall, the framework emphasizes high-fidelity simulations, multimedia content delivery, and interactive experiences, highlighting innovative approaches to enhance the quality of education and learning outcomes in healthcare.

**Specific Objective 2:** Determine the Technical Requirements Framework for the development and implementation of VPAR systems, encompassing hardware, software, and supporting technologies necessary for seamless integration and optimal learning outcomes.

Table 4 presents an overview of the essential technical requirements necessary for the effective implementation of Virtual Patient (VP) and Augmented Reality (AR) technologies in educational settings.

The technical requirements for implementing Virtual Patient (VP) and Augmented Reality (AR) technologies in healthcare education were categorized into hardware, software, and technology types.

*Hardware:*

- Handheld displays: Transparent LCD displays for AR overlays (n = 7).

- Projection screens: Matte mirrors for direct projection (n = 5).

- Combined systems: Indoor projectors with reflective materials (n = 15).

*Software:*

- Web-based systems: UT TIME, AR ticor, and Microsoft Teams Remote Assist.

- 3D imaging and simulation: CHARM Simulation, Magic Leap One, and Unity.

- Specialized tools: PalpSim for tactile feedback and Immersive Touch for surgical training.

*Technology Types:*

- Web-based systems: Virtual patient platforms (n = 27).

- Image-based systems: Marker-based (n = 7) and marker-less (n = 20) technologies.

Most studies emphasized web-based systems and image-based, marker-less technologies for virtual patients. Handheld displays and projection screens were frequently highlighted in hardware, while software components featured advanced interfaces like HoloLens and tools for 3D imaging and game development, such as Microsoft Teams Remote

**Table 4. Components of technical requirements.**

| Components | Items | Description | Studies |
|---|---|---|---|
| **Type of technology** | Web-based computer systems, | Web-based computer systems are used for virtual patient technology. | 27 |
| | Location-based systems | In the location-based group, which is dependent on mobile phones, using GPS, Wi-Fi, the location of the person is identified and the information that is stored in the system around the person is displayed to him, so that by moving The person in the environment, the information will be updated. | 5 |
| | Image-based systems | In the second image-based group, which depends on image recognition technology, the location of physical objects in the real environment is identified and digital content is added to it. This form also has two types with markers (marker based) and without markers (marker less). | 22 |
| | Marker-based technologies | In the marker type, some signs are used, and the system provides digital content by identifying them. | 7 |
| | Marker-less technologies | In the non-marker type, physical objects are identified by the system | 20 |
| **Hardware** | Handheld displays (this is transparent) | Reference Some AR systems use hand-held, flat-panel LCD displays that use an attached camera to provide vision-based augmentations via video. The handheld display acts like a window or a magnifying glass that shows real objects with an AR overlay. | 7 |
| | Projection screens (matte mirror) | In this approach, the desired virtual information is directly projected onto the physical objects for addition. At its simplest, the goal is to have the increments be flush with the surface they're projected onto, and project them from a room-mounted projector without the need for special glasses. | 5 |
| | Combined | Another approach to visual AR relies on indoor projectors, whose images are projected onto objects in the world along the viewer's line of sight. Objects of interest are covered with a reflective material that reflects light back along the angle of incidence. | 15 |
| **Software** | UT TIME | The UT Dallas Center for Modeling and Simulation/Virtual Humans and Artificial Communities Lab (Center) has conducted research on the educational effectiveness of several virtual health care simulations it has designed and developed. | [10] |
| | AR ticor | To visualize and interact with 3D holographic images and models in the medical field | [16] |
| | Microsoft teams remote assist | This software is an interface for Halogens glasses that can be transferred based on a flash memory. | [12] |
| | chariot augmented reality medical(CHARM) • Magic leap one | It integrates instant communications into a portable medical Simulation with a patient and holographic monitor. ACLS (Instructor-led Simulation) is an AR Simulation r of advanced cardiovascular life. • Augmented reality headset that you can both hear and speak and each participant directs their interaction with their hands by holographic. | [19] |
| | AR Kit2 • NACSL | Is a software toolkit for quickly building AR apps that is freely available? • Simulation for nursing cases and vital signs. | [20] |
| | Campinas | Dental Simulation mobile application suitable for Android and Apple phones. Powered by Vuforia (PTC), a parametric technology company, this software delivers an immersive experience using a mobile device's camera or VR glasses | [27] |
| | Metaio AR | The Metaio AR platform is used to develop digital artifacts and track learning contexts. The Metaio platform is based on the Augmented Reality Experience Language (AREL) and is structured on a JavaScript binding in combination with XML content. | [36] |
| | Simulation best education(SBE) | Simulation-based training that implements technical and non-technical skills, clinical judgment, review, reflection, evaluation, and psychomotor skills. | [29] |
| | star cloud | 3D reconstruction of patients' lungs using CT scan data using star cloud workstation, which is a 3D game maker program from visual 3D | [25] |
| | 3D virtual surgical planning(VSP) | Used MIR and CT to image a 3D virtual model and video screen of the patient from the VRrinder plugin. | [34] |
| | Virtual Patient Immersive Trainer (VPIT) | Is an augmented reality medical simulation trainer called Virtual Patient Immersive Trainer (VPIT)? This Augmented Reality (AR) trainer provides medical simulation training by showing a virtual patient displaying dynamic, life-like perceptual signs (such as changes in breathing, skin color) over the real world. Depending on training goals and facility resources, the VPIT system can work with or without a dummy. | [28] |

*(Continued)*

**Table 4.** (Continued)

| Components | Items | Description | Studies |
|---|---|---|---|
| | Plasm | Developed a new augmented reality simulation called Plasm that allows trainees to feel a virtual patient using their hands. The touch stage requires both force and tactile feedback | [38] |
| | Bio Harness BT | Zephyr Technology's Bio Harness BT sensor technology is used by third-party manufacturers of wearable fitness devices to add biometric monitoring capabilities. An example of this technology is Armour's E39 electronic compression undergarment, which tracks data such as the wearer's breathing and heart rate. Data can be transferred to a computer or mobile device. | [37] |
| | Immersive Touch | The Immersive Touch augmented reality system was developed at the University of Illinois at Chicago and combines immediate tactile feedback with a high-resolution stereoscopic display. An electromagnetic head tracking system provides dynamic vision as the user moves their head. A half-silvered mirror is used to create an augmented reality environment that integrates the surgeon's hands, the virtual instrument, and the virtual patient into a common workload while eliminating image occlusion. | [13] |
| | vitrea 2 | Is a fast and intuitive tool in the clinical context that includes a graphical interface and built-in automation? Free software in this field includes radiant DICOM and Dim viewer 3 | [23] |
| | VLSI | Integrated Circuit Design 1 is a full customized course on VLSI systems using Chip Wise. The VLSI system uses industry-leading electronic design automation (EDA) tools. The EDA tools used in this research include ModelSim from Model Technology, which is used for simulation, Synplify from Synplicity, which is used for synthesis, etc., and Max+plus II from Altera, which is used for placement and routing of Altera FPGAs. is used We are also evaluating Renoir from Mentor Graphics to allow graphic design to flow into our chosen top-down design flow. | [40] |
| | ZBrush<br>• unity<br>• AR enhanced piloting | The video frame was painted using it on the 3D model of Adam's head with layered animations using Aftereffects, a video software.<br>• The platform is a virtual environment and a game engine for content.<br>• Its approach is based on a scenario based on mobile phone and video. | [21] |

Assist, 3D VSP, Magic Leap One, and Unity. These platforms support simulations across healthcare fields, showcasing innovative tools designed to enhance education through immersive experiences.

Overall, the figure presented below outlines a comprehensive conceptual framework for VPAR systems. This framework was developed after integrating data and insights from the analysis of relevant literature. It provides a structured overview of the key components and their interrelationships within the VPAR ecosystem. Understanding this framework is crucial for the effective design, development, and implementation of immersive learning technologies in healthcare education.

The framework encompasses several interconnected elements, including the characteristics of VPAR design models, educational strategies, interaction and feedback mechanisms, learning environments, and the evaluation of learning outcomes. These components work together to create an engaging and effective learning experience for healthcare students and professionals. (Fig 5).

## Discussion

This scoping review provides a comprehensive overview of the Integrated Learning Design Framework and Technical Requirements for Augmented Reality (AR)-based Virtual Patient (VP) systems in healthcare professions education. The findings highlight the importance of design models, educational strategies, interaction mechanisms, learning environments, and evaluation methods in creating effective and immersive learning experiences. By drawing on established

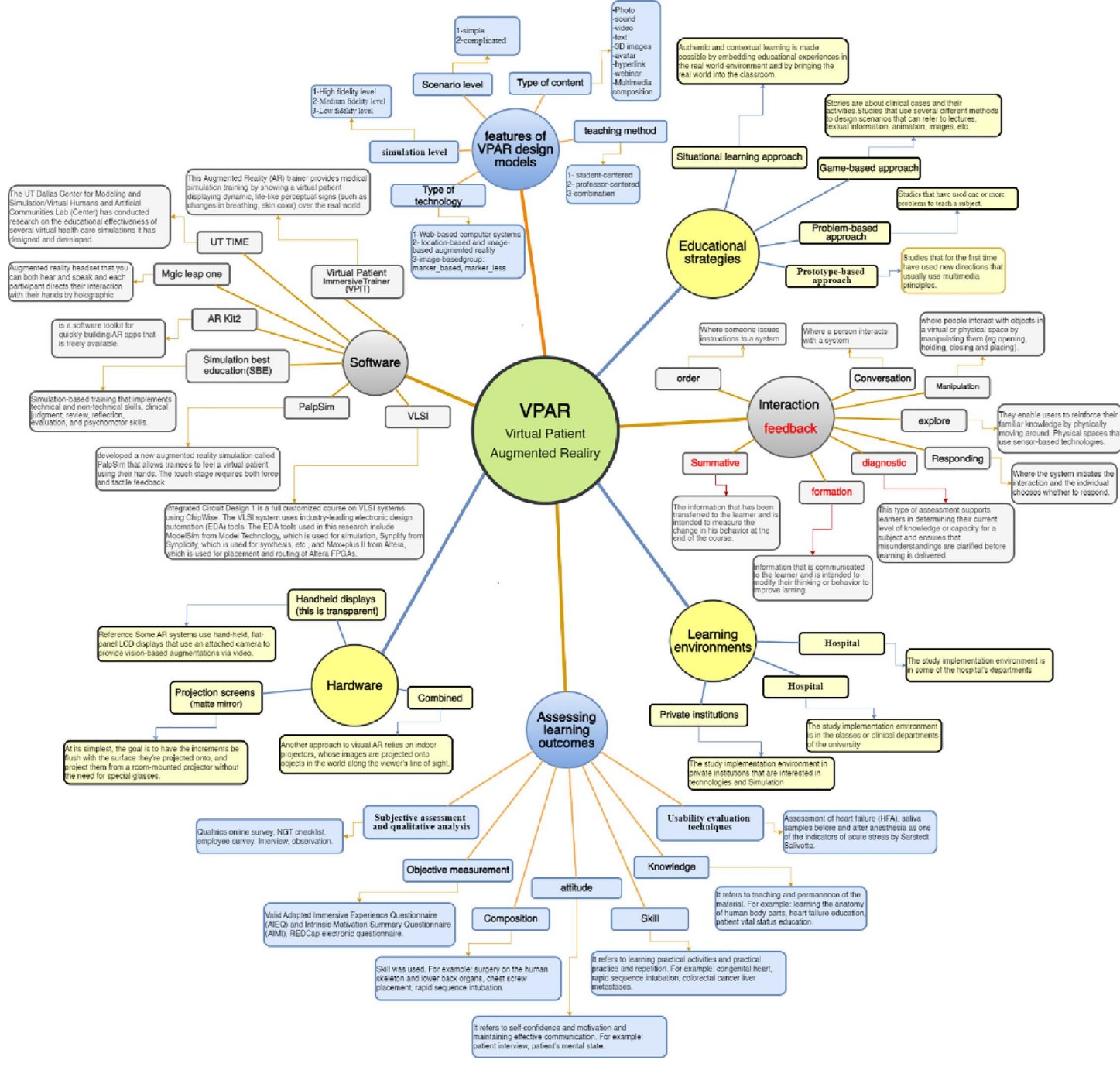

**Fig 5. Conceptual framework of virtual patient augmented reality (VPAR).**

educational models such as the ADDIE model for instructional design and the Technology Acceptance Model (TAM), this review ensures a robust foundation for evaluating VP/AR technologies [41,42].

A central theme identified across the reviewed literature is the prevalence of student-centered learning approaches, often facilitated by instructor guidance. Our analysis reveals a consistent application of pedagogical strategies like problem-based learning, prototype-based learning, situational learning, and game-based learning to enhance learner engagement and knowledge acquisition [12,43,44]. These strategies are commonly embedded within simulations of varying fidelity—high, medium, and low—leveraging a rich array of multimedia content, including audio, images, videos, 3D models, and avatars, to create realistic and interactive learning scenarios [12,21,27,28]. In terms of interaction, VP/AR systems support a diverse set of methods such as commanding, conversing, manipulating, exploring, and responding, all of which contribute to active participation and the development of essential skills [45–47]. The provision of feedback, categorized as diagnostic, formative, and summative, is consistently highlighted, with formative feedback being particularly emphasized for its role in guiding learning and improving outcomes [12,17,21].

VP/AR systems are implemented in diverse learning environments, including hospitals, universities, and private institutions, each offering unique opportunities for hands-on training and skill development [16,18,25]. Evaluation methods include objective measurements (e.g., tests, surveys) and subjective assessments (e.g., interviews, observations). Kirkpatrick's evaluation model, which assesses reaction, learning, behavior, and results, is commonly used to measure the effectiveness of VP/AR technologies [16,18,19]. The hardware for VP/AR systems ranges from handheld displays (e.g., computers, monitors) to projection displays (e.g., HoloLens, mobile cameras) [10,24,25,35], often combined with advanced software platforms such as Unity3D, ARKit2, and CHARM Simulation [21,23,27]. These technologies are evaluated using models like the Technology Acceptance Model (TAM) and the System Usability Scale (SUS), focusing on usability objectives such as effectiveness, efficiency, safety, and user satisfaction [48].

Our study both corroborates and extends existing research on VP/AR technologies in healthcare education. For example, the emphasis on location-based AR and image-based AR for achieving high-fidelity simulations, as highlighted by Moro et al. (2017) and Herbert et al. (2021) [20,47], aligns with our findings regarding the importance of realistic and contextually relevant learning experiences. However, while studies such as Aebersold et al. (2018) and Barsom et al. (2016) have demonstrated the educational benefits of AR in specific domains like anatomy learning and clinical simulations [29,49], they often present a fragmented view of the integrated potential of VP and AR. In contrast to these more focused studies, our scoping review provides a more holistic framework that synthesizes design principles, assessment strategies, and educational theories to optimize the comprehensive application of VP/AR systems in healthcare education. This broader perspective allows for a more nuanced understanding of how these technologies can be effectively leveraged across different learning contexts and for various educational objectives.

Furthermore, while previous research has touched upon the various components of VP/AR systems, this review offers a more integrated perspective by explicitly linking the technical requirements with established learning design frameworks. For instance, our analysis highlights how specific interaction mechanisms supported by AR technology can directly facilitate active learning strategies advocated by models like problem-based learning. This explicit mapping between technical capabilities and pedagogical approaches represents a significant contribution beyond studies that primarily focus on either the technical aspects or the educational outcomes in isolation.

Despite the considerable promise of VP/AR technologies, our review has identified several limitations within the current body of research. The prevalence of small sample sizes across many studies raises concerns about the generalizability of their findings. Additionally, the high costs associated with specialized AR hardware, such as the HoloLens, present significant barriers to widespread adoption [20,50]. Concerns regarding the reliability and validity of certain assessment tools underscore the need for the development of more robust and standardized evaluation methods. The novelty effect, which may temporarily inflate learner satisfaction and performance, necessitates longitudinal studies to accurately assess the long-term retention of knowledge and skills acquired through VP/AR interventions [51]. To address these limitations, future

research should prioritize conducting larger, multi-institutional studies to enhance the generalizability of findings. Exploring more cost-effective hardware solutions is crucial for promoting broader accessibility. The development and validation of comprehensive assessment tools are essential for accurately measuring the impact of VP/AR on learning outcomes. Furthermore, investigating the long-term retention of knowledge and skills and the seamless integration of VP/AR technologies into broader medical curricula are vital steps towards realizing their full educational potential.

In comparison to existing reviews in this area, our scoping review provides a more granular and integrated analysis of the technical requirements and learning design frameworks underpinning VP/AR systems. While other reviews may focus on specific applications or outcomes, our work offers a broader synthesis of the key components and considerations for designing and implementing effective VP/AR-based learning experiences. This comprehensive approach allows educators and researchers to gain a deeper understanding of the interdependencies between technical features, pedagogical strategies, and assessment methods.

In summary, this scoping review reaffirms the transformative potential of VP/AR technologies in healthcare professions education. By effectively integrating advanced technologies, established educational models, and comprehensive evaluation methods, VP/AR systems have the capacity to significantly enhance learning experiences and improve outcomes for medical students and professionals. However, our analysis underscores a critical need for more balanced evaluation methods that incorporate both formative and summative feedback, as well as continued research into immersive AR technologies that offer fully integrated and dynamic learning environments. Future research should also focus on addressing the identified limitations, particularly regarding generalizability, cost-effectiveness, assessment rigor, and the long-term impact of these technologies.

The limitations of this review, inherent to its scope and the current evidence base, suggest that our identification of the fundamental components of virtual patient and augmented reality technologies in healthcare professions education may not be exhaustive. Scoping reviews inherently involve certain limitations, as they aim to map the breadth of evidence rather than conduct detailed quality assessments of individual studies. Our search strategy, utilizing keywords such as simulation, virtual patient, augmented reality, mixed reality, extended reality, and healthcare professions education, may not have captured all relevant literature. The selection of components was based on existing studies and resources from both technological and educational perspectives, which may not comprehensively encompass all aspects. Additionally, many included studies provided overarching insights rather than in-depth specifics, potentially leading to incomplete information. However, we have prioritized transparency by providing clear decision-making criteria for our evidence compilation, believing that our findings offer a valuable overview of the current state of virtual patient and augmented reality technologies in healthcare education and lay the groundwork for future research.

## Conclusion

This scoping review examined virtual patient (VP) integrated with augmented reality (AR) in healthcare professions education. The review found VP/AR simulations to be promising for creating interactive and realistic learning experiences. Effective design requires aligning content with learning goals, balancing realism with usability, incorporating interactive features, and providing meaningful feedback. While limitations like small sample sizes and high hardware costs exist, future research can address these by conducting larger studies, exploring cost-effective hardware, and developing robust assessments. Ultimately, this review contributes to understanding VP/AR in healthcare professions education and paves the way for optimizing their effectiveness for wider adoption.

## Supporting information

**S1 File. PRISMA ScR fillable checklist.**
(DOCX)

**S2 File. Data obtained based on search strategy in 6 databases: MEDLINE (PubMed) (S1 Table), Science Direct (Elsevier) (S2 Table), Cochrane library (S3 Table) ERIC (S4 Table), Web of Science (Clarivate) (Supporting Information table 50, Scopus(S5 Table).**
(DOCX)

## Author contributions

**Conceptualization:** Farshid Chahartangi, Nahid Zarifsanaiey, Manoosh Mehrabi.

**Data curation:** Farshid Chahartangi, Nahid Zarifsanaiey.

**Formal analysis:** Farshid Chahartangi, Nahid Zarifsanaiey.

**Investigation:** Farshid Chahartangi, Nahid Zarifsanaiey.

**Methodology:** Farshid Chahartangi, Nahid Zarifsanaiey, Bahareh Zeynalzadeh Ghoochani.

**Project administration:** Nahid Zarifsanaiey.

**Supervision:** Nahid Zarifsanaiey, Manoosh Mehrabi.

**Visualization:** Bahareh Zeynalzadeh Ghoochani.

**Writing – review & editing:** Farshid Chahartangi, Nahid Zarifsanaiey, Manoosh Mehrabi.

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
