## [Decision Letter · Decision Letter 0]

PONE-D-25-10169Integrating Augmented Reality Virtual Patients into Healthcare Training : A Scoping Review of Learning Design and Technical RequirementsPLOS ONE

Dear Dr. Zarifsanaiey,

Thank you for submitting your manuscript to PLOS ONE. After careful consideration, we feel that it has merit but does not fully meet PLOS ONE’s publication criteria as it currently stands. Therefore, we invite you to submit a revised version of the manuscript that addresses the points raised during the review process.

**ACADEMIC EDITOR: ** Please address the concerns by all the reviewers before I consider accepting your manuscript. 

We look forward to receiving your revised manuscript.

Kind regards,

Ziyu Qi

Academic Editor

PLOS ONE

Journal Requirements:

2. In the online submission form, you indicated that the data supporting this study's findings are available from the corresponding author on request.

Additional Editor Comments :

Please address the concerns by all the reviewers before I consider accepting your manuscript.

Reviewers' comments:

Reviewer's Responses to Questions

**Comments to the Author**

1. Is the manuscript technically sound, and do the data support the conclusions?

Reviewer #1: Yes

Reviewer #2: Yes

Reviewer #3: Yes

2. Has the statistical analysis been performed appropriately and rigorously? 

Reviewer #1: Yes

Reviewer #2: Yes

Reviewer #3: Yes

3. Have the authors made all data underlying the findings in their manuscript fully available?

Reviewer #1: Yes

Reviewer #2: Yes

Reviewer #3: Yes

4. Is the manuscript presented in an intelligible fashion and written in standard English?

Reviewer #1: Yes

Reviewer #2: Yes

Reviewer #3: Yes

5. Review Comments to the Author

Reviewer #1: Augmented reality and virtual patients offer interactive and immersive learning experiences, with AR enhancing the understanding of complex concepts and VPs providing hands-on practice in clinical scenarios. In this well-written review, the authors have done a nice job to summarize the relavent literatur. This review may be helpful for the readers to get a quick picture of this popular topic.

Reviewer #2: Manuscript Number: PONE-D-25-10169

Article Type: Research Article

Title: Integrating Augmented Reality Virtual Patients into Healthcare Training : A Scoping

Review of Learning Design and Technical Requirements

Dear Sir/madam

Thank you for your efforts, and for considering our journal for publication of your work.

Title

Title is concise and reflect the study content properly.

Abstract

- Abbreviations appearing for the first time and not explained before (like; AR, VP,..). authors need to make sure these are established as the beginning before continue using it. For example augmented reality (AR) at the start.

- As a review article your database you carried the search on should be included in full rather than etc.

- A summary of inclusion/ exclusion criteria is needed, how you filtered 924 into 27? (no details needed)

Methods

- Clearly state if studies using only one technology (VP) or (AR) were included in the review, and if so does your objectives restrict the study to inclusion of those using both technologies only?

- The time frame of the included studies need to be clarified? Was it from specific date (ex; 2000) or open to any available literature?

Results

- Results are well structured and represent the objectives.

- Authors need to explain the justification behind including IT students? Exclusion criteria mentioned studies not conducted at healthcare level.

- The paragraph headed “Medical Specialties and Educational Focus”, and the sub “Overview of Medical Specialties and Educational Contexts” are repetitive with high redundancy. Authors are advised to draft this section.

- Authors also need to explain how these classifications were made? Why stroke diagnosis were put with “therapeutic skills” category? Was there any method/ software of qualitative analysis used to draw these data?

Discussion

- The discussion section started by re-stating the aim and the main finding which is good.

- However, detailed data were presented. Authors need to focus on bringing more comparison with the existing literature rather than detailing the same data presented at the results section.

- Limitations were clearly stated, authors may as well explain what has been done to address these limitations and its potential effect on the presented results.

Conclusion

- Conclusion represent the stated objectives and data. Future implications and recommendations were presented.

Reviewer #3: The authors reference PRISMA-ScR and JBI guidelines and adopt a PCC framework, which is appropriate.

The manuscript refers to a “systematic scoring process” for evaluating included studies (p.6); however, no validated appraisal tool is cited. In “Data collection process”, clarify whether the authors formally appraised the quality of the included studies, and if so, to:

Name the tool used (e.g., JBI checklist),

Explain the process, and

Justify its use in the context of a scoping review.

6. PLOS authors have the option to publish the peer review history of their article (what does this mean? ). If published, this will include your full peer review and any attached files.

**Do you want your identity to be public for this peer review?** For information about this choice, including consent withdrawal, please see our Privacy Policy .

Reviewer #1: **Yes: ** Xiaolei Chen

Reviewer #2: No

Reviewer #3: No

---

## [Author Response · Author response to Decision Letter 1]

21 Apr 2025

Dear Esteemed Reviewers of PLOS ONE,

I am pleased to resubmit the revised manuscript titled “Integrating Virtual Patients and Augmented Reality in Healthcare Professional Education: A Scoping Review of Learning Design and Technical Requirements.” I sincerely appreciate the constructive feedback from the editor and reviewers, which has significantly enhanced the quality of this article. We have made several positive changes based on your suggestions, and the revised manuscript includes highlighted modifications for your review. I would like to express my profound gratitude for the time and attention you dedicated to this review. Your insightful critiques have been invaluable in improving the structure, content, and scientific rigor of our research.

Additionally, I extend my heartfelt thanks to Dr. Ziyu Qi, the Academic Editor of PLOS ONE, for his diligent follow-up and for the opportunity to publish in this esteemed journal.

In the following sections, we will address each of the reviewers' comments in detail and outline the corresponding revisions made in the manuscript.

Sincerely yours,

Dr. Nahid Zarifsaniey

Reviewer #1:

We extend our sincere gratitude for your kind consideration and positive feedback regarding the present study. Receiving such encouraging and affirmative responses, particularly your assessment of the research as beneficial, effective, and possessing scientific merit, is a source of considerable satisfaction and reinforces our motivation.

Your encouragement and validation significantly strengthen our resolve to pursue high-quality research and contribute to the advancement of knowledge in this field. We deeply appreciate the meticulous attention and time you dedicated to the thorough review of our manuscript, and we are highly gratified that our work has met with your approval and endorsement.

Reviewer #2:

We would like to express our sincerest gratitude and deepest appreciation for your valuable insights and profound understanding of this manuscript. Your meticulous observations and expert guidance have not only clearly illuminated the strengths and weaknesses of our study but have also paved a clear path for its improvement and the enhancement of its scientific quality.

We firmly believe that you’re constructive and precise suggestions have significantly contributed to the enrichment of the content and the strengthening of the methodological rigor of our research, transforming it into a more robust and scientifically sound work. The positive impact of your invaluable feedback will be clearly evident in the revised version of the manuscript.

Abstract

- Abbreviations appearing for the first time and not explained before (like; AR, VP). Authors need to make sure these are established as the beginning before continue using it. For example augmented reality (AR) at the start.

Response: Your valuable comment was taken into consideration and added to the text of the article.

Text added to the article abstract:

"Augmented reality (AR) enables users to view the real world with enhanced digital information, making it a transformative tool in education. Virtual patients (VPs) technology is also defined as “a specific type of computer-based application that simulates real-world clinical scenario. AR and VPs offer interactive and immersive learning experiences, with AR enhancing the understanding of complex concepts and VPs providing hands-on practice in clinical scenario."

- As a review article your database you carried the search on should be included in full rather than etc.

Response: Your valuable comment was accepted and this section was fully stated.

Text added to the article abstract:

"MEDLINE (PubMed), Science Direct (Elsevier), Web of Science (Clarivate), Cochrane library, ERIC, Scopus."

- A summary of inclusion/ exclusion criteria is needed, how you filtered 924 into 27? (no details needed).

Response: Your suggestion has been incorporated, and the necessary corrections are now present in the manuscript. To ensure a rigorous selection process aligned with the study's purpose and specified criteria, two researchers independently performed the abstraction and full-text review of the identified studies. A third researcher then conducted an independent verification of the extracted information, which resulted in the final set of articles.

Text added to the article abstract:

"A comprehensive search yielded 924 potential studies. Articles were selected via a two-stage screening process, involving title/abstract and full-text reviews based on predefined inclusion criteria. Disagreements were resolved through consultation, resulting in 27 studies being included."

Methods

- Clearly state if studies using only one technology (VP) or (AR) were included in the review, and if so does your objectives restrict the study to inclusion of those using both technologies only?

Response: We selected articles that integrated both virtual patient and augmented reality technologies. The selection of studies involved identifying articles with relevant keywords in their title or abstract. However, for example, in some articles where the term "virtual patient" was absent from their title, but upon review of the abstract and full text, the use of virtual patient simulation (such as: brain tumors and heart problems, etc.) was evident and aligned with the objective of the present study.

- The time frame of the included studies need to be clarified? Was it from specific date (ex; 2000) or open to any available literature?

Response: No date limitations were applied in this study, resulting in an open literature review. The absence of a date filter was a deliberate choice to allow for the inclusion of all pertinent studies.

Inclusion Criteria:

Study Focus: Only studies integrating both augmented reality (AR) and virtual patient (VP) technologies in healthcare professions education were included. Studies utilizing only one technology (AR or VP alone) were excluded unless they explicitly combined both.

Study Designs: Observational (e.g., cohort studies), quasi-experimental (e.g., randomized controlled trials, pre-post studies), and descriptive (e.g., case studies, qualitative studies).

Language: English-language publications.

Timeframe: No date restrictions were applied; all available literature meeting the criteria was considered.

Exclusion Criteria:

• Grey literature, non-full-text articles, studies not directly addressing AR/VP integration, and publications from dubious databases.

• Studies focused solely on virtual reality (VR) without AR or VP components.

The PCC framework

Population: Healthcare profession students (medical, nursing, allied health) engaged in AR-based VP educational interventions.

Concept: Integration of AR and VP as educational tools, including:

Design, implementation, and evaluation of AR-based VP simulations.

Impact on learning outcomes (e.g., knowledge acquisition, clinical reasoning).

Context: Healthcare education settings (undergraduate, postgraduate, continuing professional development).

• Population (participants): Healthcare profession students (e.g., medical, nursing, and…).

Results

- Authors need to explain the justification behind including IT students? Exclusion criteria mentioned studies not conducted at healthcare level.

Response:

We sincerely appreciate the reviewer’s valuable feedback. We corrected the terminology from "IT students" to "medical informatics students" throughout the manuscript to accurately reflect their role in healthcare technology development.

- The paragraph headed “Medical Specialties and Educational Focus”, and the sub “Overview of Medical Specialties and Educational Contexts” are repetitive with high redundancy. Authors are advised to draft this section.

Response:

Thank you for your constructive feedback regarding redundancy in the sections titled “Medical Specialties and Educational Applications” and “Overview of Medical Specialties and Educational Contexts.” We have revised these sections to reduce repetition and enhance clarity, ensuring that the content is more concise and focused.

Medical Specialties and Educational Applications

The included studies (n=27) showcased the application of augmented reality-based virtual patient (AR-VP) technology across various medical specialties. Two reviewers independently extracted and classified the data according to the primary educational focus of each study, resolving discrepancies through discussion and consulting a third reviewer when necessary. The classifications were derived inductively from study objectives and outcomes, without utilizing qualitative analysis software.

The major themes and applications identified included:

• Advanced Clinical Procedures: Examples include mpMRI-guided prostate biopsies, emergency cardiac arrest management, and treatment planning for colorectal cancer metastases.

• Anatomical and Physiological Education: Focus areas encompass female breast anatomy, lower limb structures, skeletal system, and congenital heart abnormalities, alongside the use of CT scans for pulmonary lesion assessment and heart physiology.

• Diagnostic and Therapeutic Training: This includes diagnostic skills such as stroke diagnosis and critical patient assessment, as well as therapeutic skills like brain tumor management, prescription writing, and patient communication. Stroke diagnosis was classified under "therapeutic skills" when studies emphasized treatment decision-making.

• Practical Clinical Skills: Hands-on training involved vital sign monitoring, chest screw placement, and heart rate interpretation.

• These studies underscore the importance of comprehensive training across medical specialties and the critical role of immersive technologies in developing essential clinical skills.

Overview of Medical Contexts

The reviewed studies covered a wide range of medical specialties and educational contexts, including:

• Advanced Procedures: mpMRI-guided prostate biopsies, emergency cardiac arrest responses, and colorectal cancer management.

• Anatomical Education: Female breast, lower limb, and skeletal anatomy.

• Diagnostic Training: Heart anatomy, physiology, pulmonary lesions, and radiological techniques (e.g., CT scans).

• Therapeutic Skills: Stroke diagnosis, brain tumor management, and patient communication.

• Practical Skills: Heart rate monitoring, chest screw placement, prescription writing, and vital signs management.

- Authors also need to explain how these classifications were made? Why stroke diagnosis were put with “therapeutic skills” category?

Response:

Thank you for your valuable feedback and insightful questions regarding our classification process.

The classifications in our study were made through a comprehensive review of the literature. Two reviewers independently extracted and categorized data based on the primary educational focus of each study. Any discrepancies between the reviewers were resolved through discussion, and a third reviewer was consulted when necessary to ensure accuracy and consensus. Stroke diagnosis was categorized under "therapeutic skills" because many studies emphasized its role in treatment decision-making. This classification reflects the importance of integrating diagnostic skills with therapeutic interventions, particularly in acute scenarios where prompt treatment is essential.

Was there any method/ software of qualitative analysis used to draw these data?

Response:

To ensure a comprehensive evaluation of the studies included in this scoping review on AR-based virtual patients in healthcare training, two independent reviewers conducted a thorough quality assessment. Each reviewer systematically appraised the studies against predefined criteria, focusing on several key areas: methodological quality, relevance to learning design and technical requirements, and alignment with the objectives of the review.

To address any initial disagreements in assessments, the reviewers engaged in iterative discussions, ultimately reaching a consensus. This collaborative approach guaranteed an unbiased and standardized selection process:

To ensure a comprehensive evaluation of the studies included in this scoping review on AR-based virtual patients in healthcare training, two independent reviewers conducted a thorough quality assessment using the JBI checklist. Each reviewer systematically appraised the studies against predefined criteria, focusing on several key areas: methodological quality, relevance to learning design and technical requirements, and alignment with the objectives of the review.

To address any initial disagreements in assessments, the reviewers engaged in iterative discussions, ultimately reaching a consensus. This collaborative approach guaranteed an unbiased and standardized selection process.

A structured scoring system was utilized to evaluate critical aspects of each study, including:

Research Design Validity: Assessing the robustness of the study's design.

Technical Implementation Rigor: Evaluating the thoroughness of the technical execution.

Educational Outcomes Measurement: Analyzing how effectively educational outcomes were measured.

Reported Limitations: Reviewing the transparency and acknowledgment of limitations within each study.

The reviewers meticulously documented their scoring rationales to enhance transparency, creating an auditable trail for methodological decisions. This rigorous approach prioritized studies that effectively addressed the integration of AR virtual patients in healthcare education while upholding high scientific standards.

Collating and Summarizing Data

Following the quality assessment phase, the data were systematically categorized and analyzed. A summary table was created to outline the characteristics and findings of the included articles, and a comprehensive list of studies was compiled. An overview of the studies was conducted by systematically analyzing their geographic distribution, publication years, outcomes, and content to identify the benefits, effects, and challenges associated with the study's objectives.

Discussion

- However, detailed data were presented. Authors need to focus on bringing more comparison with the existing literature rather than detailing the same data presented at the results section.

Response: We extend our sincere gratitude for your meticulous suggestion. Considering that this is a scoping review study, our initial approach involved specifying the findings and subsequently comparing them with other studies based on these identified results. The body of literature employing the integration of these technologies was limited, often providing scant detail. Nevertheless, the research team endeavored to discuss those articles demonstrating the highest relevance to the study's objective. Notwithstanding these limitations, we sought to examine the comparisons with due attention to the specific details presented within the articles.

Text added to the article abstract:

"A central theme identified across the reviewed literature is the prevalence of student-centered learning approaches, often facilitated by instructor guidance. Our analysis reveals a consistent application of pedagogical strategies like problem-based learning, prototype-based learning, situational learning, and game-based learning to enhance learner engagement and knowledge acquisition (12, 43, and 44). These strategies are commonly embedded within simulations of varying fidelity—high, medium, and low—leveraging a rich array of multimedia content, including audio, images, videos, 3D models, and avatars, to create realistic and interactive learning scenarios (12, 21, 27, 28). In terms of interaction, VP/AR systems support a diverse set of methods such as commanding, conversing, manipulating, exploring, and responding, all of which contribute to active participation and the development of essential skills (45-47). The provision of feedback, categorized as diagnostic, formative, and summative, is consistently highlighted, with formative feedback being particularly emphasized for its role in guiding learning and improving outcomes (12, 17, and 21).

Our study both corroborates and extends existing research on VP/AR technologies in healthcare education. For example, the emphasis on location-based AR

---

## [Editor Report · Decision Letter 1]

Integrating Augmented Reality Virtual Patients into Healthcare Training : A Scoping Review of Learning Design and Technical Requirements

PONE-D-25-10169R1

Dear Dr. Zarifsanaiey,

We’re pleased to inform you that your manuscript has been judged scientifically suitable for publication and will be formally accepted for publication once it meets all outstanding technical requirements.

Kind regards,

Ziyu Qi

Academic Editor

PLOS ONE
---

## [Editor Report · Acceptance letter]

PONE-D-25-10169R1

PLOS ONE

Dear Dr. Zarifsanaiey,

I'm pleased to inform you that your manuscript has been deemed suitable for publication in PLOS ONE. Congratulations! Your manuscript is now being handed over to our production team.

Kind regards,

on behalf of

Mr. Ziyu Qi

Academic Editor

PLOS ONE